# Maternal Allergy and the Presence of Nonhuman Proteinaceous Molecules in Human Milk

**DOI:** 10.3390/nu12041169

**Published:** 2020-04-22

**Authors:** Pieter M. Dekker, Sjef Boeren, Alet H. Wijga, Gerard H. Koppelman, Jacques J. M. Vervoort, Kasper A. Hettinga

**Affiliations:** 1Food Quality and Design Group, Wageningen University & Research, Bornse Weilanden 9, 6708 WG Wageningen, The Netherlands; pieter.dekker@wur.nl; 2Laboratory of Biochemistry, Wageningen University & Research, Stippeneng 4, 6708 WE Wageningen, The Netherlands; sjef.boeren@wur.nl (S.B.); jacques.vervoort@wur.nl (J.J.M.V.); 3National Institute of Public Health and the Environment—Centre for Prevention and Health Services Research, Antonie van Leeuwenhoeklaan 9, 3721 MA Bilthoven, The Netherlands; alet.wijga@rivm.nl; 4Beatrix Children’s Hospital, Department of Pediatric Pulmonology and Pediatric Allergology, University of Groningen, University Medical Centre Groningen, Hanzeplein 1, 9713 GZ Groningen, The Netherlands; g.h.koppelman@umcg.nl; 5Groningen Research Institute for Asthma and COPD (GRIAC), University of Groningen, University Medical Centre Groningen, Hanzeplein 1, 9700 RB Groningen, The Netherlands

**Keywords:** allergen transfer, human milk, β-lactoglobulin, maternal allergy, nonhuman proteins

## Abstract

Human milk contains proteins and/or protein fragments that originate from nonhuman organisms. These proteinaceous molecules, of which the secretion might be related to the mother’s allergy status, could be involved in the development of the immune system of the infant. This may lead, for example, to sensitization or the induction of allergen-specific tolerance. The aim of this study was to investigate the relation between maternal allergy and the levels of nonhuman proteinaceous molecules in their milk. In this study, we analysed trypsin-digested human milk serum proteins of 10 allergic mothers and 10 nonallergic mothers. A search was carried out to identify peptide sequences originating from bovine or other allergenic proteins. Several methods were applied to confirm the identification of these sequences, and the differences between both groups were investigated. Out of the 78 identified nonhuman peptide sequences, 62 sequences matched *Bos taurus* proteins. Eight peptide sequences of bovine β-lactoglobulin had significantly higher levels in milk from allergic mothers than in milk from nonallergic mothers. Dietary bovine β-lactoglobulin may be absorbed through the intestinal barrier and secreted into human milk. This seems to be significantly higher in allergic mothers and might have consequences for the development of the immune system of their breastfed infant.

## 1. Introduction

The human milk proteome comprises more than once thought. Besides a vast number of human proteins and peptides, it also includes nonhuman intact proteins, large protein fragments, and peptides (later referred to as proteinaceous molecules). The presence of such molecules in human milk has been demonstrated decades ago with immunochemical analysis [1] and has recently been confirmed with mass spectrometry [2].

According to studies using mass spectrometry, the main biological source of the nonhuman proteinaceous molecules in human milk seems to be bovine milk. A peptidomics study demonstrated the presence of two peptides originating from bovine β-lactoglobulin (BLG) and one originating from α_S1_-casein [3]. In a later study, this was extended with peptides from α-lactalbumin (ALA), κ-casein, β-casein, and lactoferrin [4]. Evidence for the presence of intact bovine caseins and BLG has recently been provided [2,4,5]. In addition to the bovine proteins and peptides, Zhu et al. also identified several peptide sequences originating from other nonhuman species, which may include allergens [2]. So far, only peanut allergen has been identified with high sequence coverage by liquid chromatography tandem mass spectrometry (LC-MS/MS) [6]. The presence of egg, wheat, and house dust mite (HDM) allergens in human milk, which has been demonstrated using immunochemical methods [7,8,9], has not been confirmed yet with LC-MS/MS analysis.

The presence of these nonhuman proteinaceous molecules in human milk raises the question of how they end up there. It has been suggested that dietary proteins can be transferred through the intestinal barrier by both paracellular and transcellular pathways [10]. Furthermore, nonhuman proteins present in the mother’s blood might also be transferred by these pathways through the mammary epithelia into the milk [11,12]. Nevertheless, it remains unclear which pathways are used for the transfer of nonhuman proteinaceous molecules into human milk.

Several of the studies that identified nonhuman proteinaceous molecules in human milk report a large interindividual variation in the levels of these molecules. An explanation for this variation has not been found yet, but it does not seem to be related to dietary intake [13]. Maternal asthma and allergy could be an important factor in this variation, since it is known that e.g., atopic eczema and asthma can have an influence on intestinal barrier integrity [14,15,16]. This could then lead to an increased passage of dietary proteinaceous molecules through the intestinal barrier. Research to date has not yet considered the relation between maternal allergic diseases and the levels of nonhuman proteinaceous molecules in human milk using LC-MS/MS. The purpose of the present study was therefore to identify nonhuman proteinaceous molecules in human milk and to investigate if the levels of these molecules were related to maternal allergy. This could be useful for further research into the mechanisms responsible for the transfer of these molecules and the effect of these molecules on the infant’s immune system.

## 2. Materials and Methods

### 2.1. Milk Samples

We used data from a population-based Dutch birth cohort study: the Prevention and Incidence of Asthma and Mite Allergy (PIAMA) Study. Details of the cohort study are described elsewhere [17,18]. In short, pregnant women were recruited from the general population during their first antenatal visit. Their children (*n* = 3963) were born in 1996/1997. Pregnant women were identified as allergic or nonallergic through a screening questionnaire. House dust samples and breastmilk samples were collected in a subgroup of the population around the child’s age of three months. Breastmilk collection was done by manual pressure or by use of a breast pump. Samples were stored in small plastic cups at −80 °C. Along with these samples, cat ownership and the frequency of consumption of milk and milk products by the mother was assessed using a questionnaire (Table 1). Maternal blood samples were collected at the child’s age of one year. The study was performed in accordance with the ethical principles for medical research involving human subjects outlined in the Declaration of Helsinki. Therefore, the study protocol was approved by the Medical Ethics Committees of the participating institutes (Rotterdam MEC 132.636/1994/39 and 137.326/1994/130; Groningen MEC 94/08/92; and Utrecht, MEC-TNO oordeel 95/50). All parents gave written informed consent.

The current study is based on a data-dependent LC-MS/MS proteomics data set that was obtained in a previous study [19]. It comprises mass spectrometry data of human milk serum protein samples from 10 allergic mothers and 10 nonallergic mothers from the cohort study. The number of mothers included is based on a power calculation, aiming at finding a 5-fold difference, as detailed in Hettinga et al. [19]. The selection of the allergic mothers was based on (a) self-reported (history of) asthma, current hay fever, current allergy for pets, or current allergy for house dust or house dust mite in combination with (b) a high level of specific IgE against HDM (≥3.50 kU/L) and (c) exposure to HDM allergen in mattress dust ((Der p 1 + Der f 1) > 600 ng/m^2^) (see Table 1). The selection of nonallergic mothers did not report any allergies or asthma. This group consisted of mothers with exposure to HDM allergen in mattress dust in the same range as the allergic mothers (600–2500 ng/m^2^) as well as mothers with much higher exposures (≥24,000 ng/m^2^). The nonallergic mothers were not tested for specific IgE against house dust mite.

From the milk samples, milk serum was obtained and serum proteins were prepared for analysis with filter-aided sample preparation. In short, full scan FTMS spectra were obtained (m/z 380 to 1400) in positive mode on an LTQ-Orbitrap system (Thermo electron, San Jose, CA, USA). The four multiply-charged peaks with the highest intensity were recorded in the linear trap in data-dependent mode (MSMS threshold: 5000). Further details of the sample preparation and LC-MS/MS analysis have been described before [19].

The data underlying the findings presented in this paper are available on request. Requests can be submitted to the PIAMA principal investigators. Their names and e-mail addresses are listed on the PIAMA website (http://piama.iras.uu.nl/index-en.php#collaboration).

### 2.2. Methods

#### 2.2.1. Data Analysis

The raw MS/MS data was analysed using the Andromeda search engine of the MaxQuant software v1.6.1.0 [20]. Since the use of large databases in proteomic data analysis affects the sensitivity of the search [21], a complete but concise database was created for this study. This database contained human milk proteins (*n* = 2569), bovine milk proteins (*n* = 1006), and allergen proteins (*n* = 721). This database is provided in the Supplementary Materials, the fasta database. Allergens were added to the database because of their immunological relevance and bovine milk proteins because the majority of the nonhuman proteinaceous molecules in human milk was previously shown to originate from bovine milk [2]. The selection of human and bovine milk proteins was made based on previous data analysis of human and bovine milk protein samples (data not published) using databases with all human or bovine proteins available in UniProtKB (both downloaded from UniProt on 16-10-2018). This was complemented with data from reviews on the bovine milk and human milk proteome [22,23]. Allergen protein sequences were obtained from UniProt on 16-10-2018 by performing a search on all proteins annotated as allergen (search term: “annotation:(type:allergen)”).

The search for peptide sequences was performed three times, in which the protein database was in silico digested with trypsin digestion, semi-specific trypsin digestion, or unspecific digestion. Maximum missed cleavages was set to two in the trypsin digestion mode. In all searches, a fixed modification was set to carbamidomethylation of cysteine. Variable modifications were set to acetylation of the peptide N-term, deamidation of the side chains of asparagine and glutamine, and oxidation of methionine, with a maximum of five modifications per peptide. The identified peptides were quantified using label-free quantification (LFQ). At both the peptide and protein levels, a false discovery rate of 1% was used. The peptide length was set from 6 to 35 amino acids. The precursor mass tolerance was set to 20 ppm, and fragment mass tolerance was set to 0.5 Da. Recalibration was carried out using a first search with a database containing common contaminants.

To remove all identifications that belong to sequences originating from human proteins, the MaxQuant output was subjected to a filtering consisting of six steps. First, all sequences originating from trypsin and keratin were removed as contaminants. Second, the reverse sequences from the decoy database were removed. Third, all sequences that had a full match with the human proteome were removed. Fourth, we removed all MS/MS scans that had a match in a separate search using only the whole human proteome database. Fifth, all sequences with an Andromeda score lower than 80 were removed to exclude low quality peptide spectrum matches (PSM). Sixth, PSMs with a second-best match to a human peptide sequence and an Andromeda score difference of <5 were removed.

#### 2.2.2. Annotation

Protein entries containing an exact match with the identified and selected peptides were found using the Peptide Match service of the online Protein Information Resource [24]. This service makes use of an up-to-date UniProtKB database. Peptides were matched to this database without isoforms, where leucine and isoleucine were treated as equivalent.

Protein and organism annotation was added using a frequency of occurrence. All matching proteins and their corresponding taxonomic lineage were listed. A leading protein was selected for each peptide sequence based on the frequency of occurrence of this protein in the peptide match results. After this, a similar approach was used on the level of taxonomy, leaving the organisms with the highest number of matches to the identified peptides as leading organism or, in case of multiple organisms, the lowest common ancestor (LCA). With this approach, *Bos taurus* was preferred over e.g., *Bos mutus* as the leading organism because of a higher number of identified peptides that matched the *Bos taurus* proteins.

#### 2.2.3. Statistical Analysis

Data analysis was carried out, and figures were made using R version 3.6.0 [25]. Missing values of LFQ intensities for the identified and selected peptide sequences were associated with levels below detection limit. Therefore, imputation was applied to ^10^log transformed LFQ intensities, with values from a normal distribution downshifted from the sample mean with 1.8 and with a standard deviation of 0.3.

Differences between the allergic and nonallergic group were tested using a two-sided unequal variances *t*-test and a Benjamini–Hochberg correction was applied on the resulting *p*-values. Significantly different peptides were selected with a *p*-value < 0.01. An additional threshold of 0.75 was set on the difference between the means of the sample groups (^10^log transformed intensity values) in order to select only significant sequences with a large between-group difference.

#### 2.2.4. Confirmatory Analysis

Bovine caseinate (prepared in-house), lactoferrin, BLG, ALA and bovine serum albumin (BSA) (Sigma-Aldrich, St. Louis, MO, USA) were dissolved in a 100 mM Tris solution and digested with trypsin. For confirmation of the nonhuman, non-bovine peptides, 12 peptides were acquired through synthesis by Royobiotech Co., Ltd. (Shanghai, China). Protein digests and synthetic peptides were analysed one by one on the same LC-MS/MS system and with the same parameters as used for the analysis of the human milk samples [19]. A summary of the workflow and confirmation of MSMS spectra is visualised in Figure 1.

## 3. Results

In this study, data-dependent shotgun proteomics data of human milk serum from 10 allergic and 10 nonallergic mothers was analysed. In a search for nonhuman proteins and protein fragments, the identified peptides were filtered and LFQ data was used for quantification.

### 3.1. Identification of Exogenous Peptides

Trypsin-digested human milk serum protein data was analysed using a database containing human milk, bovine milk, and allergen protein sequences. The identified peptide sequences were filtered to remove all human peptides and as many false positives as possible. In total, 78 nonhuman peptide sequences were identified in 20 samples. From these, 62 sequences had *Bos taurus* as leading organism (Table 2) and 16 sequences were assigned to non-bovine allergens (Table 3). Most of the identified peptide sequences (*n* = 48) were from trypsin-digested proteins. In addition, 10 peptides were semi-trypsin digested and 20 were not digested by trypsin.

Peptide sequences of 29 different bovine proteins were identified. From these proteins, the major bovine milk allergen, BLG, was identified with the highest sequence coverage (67%). To confirm the identification of the bovine sequences, tryptic digests of the major bovine milk proteins, BLG, BSA, α_S1_-casein, and ALA were analysed. This led to confirmation of 20 sequences based on MS/MS spectrum and retention time. The identification of another 16 sequences was confirmed by MS/MS spectra and retention times of these sequences in a bovine milk protein data set (data set not published). The protein with the second highest sequence coverage is bovine serum albumin (BSA). The identified peptide sequences (*n* = 14) correspond to a sequence coverage of (22%). In contrast to studies from other groups that removed these peptides from their data sets because of the use of BSA as quality control in their studies [2,4], we did not remove these peptides from our results. No evidence was found for carryover of BSA peptides in the LC system. Several BSA peptide sequences identified in human milk were not found in a trypsin-digested BSA standard solution that was used in our laboratory, indicating that the BSA-derived peptides are genuine. Considering these findings, it is likely that BSA or its proteolytic fragments are present in human milk.

From the non-bovine allergen proteins or protein fragments, proteins from *Felis catus* (domestic cat), *Equus caballus* (horse), and *Triticum aestivum* (common wheat) were identified with two or more peptide sequences. To confirm the identification of these peptide sequences, one synthesized sequence of each identified protein was acquired and analysed. From these nine peptides, two sequences were confirmed based on MS/MS spectrum and retention time. These sequences had cat and horse serum albumin as leading protein. The remaining seven sequences could not be confirmed. In several cases, the PSM of the synthesized peptide resembled the PSM of the human milk samples, but the retention time differed significantly. These PSMs are likely false positives, showing that the search for low abundant peptide sequences in shotgun proteomics is prone to finding PSMs with artefacts or co-eluted peptides. This confirms the importance of the used method in which synthesized peptides were used for confirmation.

In addition to the 16 sequences that were assigned to non-bovine allergens, 11 peptides were annotated with *Hevea brasiliensis* (Rubber tree) as leading organism. Two of these sequences were confirmed with MS/MS spectra and retention time of synthesized peptides. Nevertheless, data analysis of three other data sets of human milk shotgun proteomics did not show the presence of these peptides (data not published). Therefore, these sequences were considered as contaminant and removed from the results.

Another possible source of false-positive identifications could be the presence of unknown human protein variants due to point mutations. Out of the 78 identified nonhuman peptide sequences, 26 have one amino acid different from their human homologue. As an example, the sequence LVNELTEFAK with *Bos taurus* as leading organism has LVNEVTEFAK as homologue in *Homo sapiens*. The V → L could therefore be the result of a point mutation. Nevertheless, for all these 26 sequences, no research was found that confirmed the occurrence of these point mutations in *Homo sapiens*.

MS/MS spectra of the identified peptides and their confirmation can be found in the Appendix A.

### 3.2. Differences between Allergic and Nonallergic Mothers

Out of the 78 nonhuman peptide sequences, 15 sequences were only identified in milk from allergic mothers, whereas in milk from nonallergic mothers, 11 unique sequences were identified. This difference can be largely attributed to sequences that match to bovine proteins (Table 2). After imputation of the LFQ data and performing a *t*-test with maternal allergy as grouping variable, 16 peptide sequences appeared to be significantly different in intensity between the two groups (Figure 2).

As shown in Figure 2, nine sequences were found to be significantly higher in intensity in milk from allergic mothers. These sequences were annotated to BLG (*n* = 8) and alpha-2-HS-glycoprotein (*n* = 1) as leading protein. The seven sequences that were significantly higher in intensity in milk from nonallergic mothers were annotated to BSA (*n* = 6) and to an uncharacterized protein (*n* = 1), with semi-specific trypsin digestion. All the significantly different sequences were annotated to proteins that originate from *Bos taurus*. As can be seen in Figure 3, there is a consistent difference between the two groups, indicating that the significant differences are not caused by outliers.

## 4. Discussion

The goal of this study was to identify nonhuman proteinaceous molecules in human milk and to investigate differences in these molecules between milk from allergic and nonallergic mothers. Out of the 78 resulting nonhuman peptide sequences identified in this study, 11 sequences were reported previously in human milk studies using LC-MS/MS [2,5]. Contrary to these studies, we focused on milk serum, discarding the caseins by ultracentrifugation. This could explain the major difference with Zhu et al. when it comes to the number of identified sequences matching with bovine caseins [2]. The relatively high levels of BLG peptide sequences that we found in milk from allergic mothers explains the high sequence coverage of BLG in the current study when compared to Zhu et al. [2]. Because we removed small peptides by filter-aided sample preparation (10–20 kDa cutoff), no comparison could be made with previous peptidomics studies [3,4]. Other qualitative differences with these previous studies can be attributed to the more strict filtering on false positives that we applied, the inclusion of serum albumins, the inclusion of semi-trypsin and non-trypsin-digested sequences, and to the inclusion of milk from allergic mothers.

The transfer of proteinaceous molecules from the mother’s intestinal tract to the mammary gland is still poorly understood, especially when it comes to intact proteins or large protein fragments. In the current study, trypsin was used to digest the proteins before analysis. It can be expected that the majority of the identified peptides was digested by trypsin. Nevertheless, some peptide sequences were identified that were not, or partly, digested by trypsin. This might indicate that there are also nonhuman protein fragments present in human milk that were digested by other proteases than trypsin, probably before these fragments even entered the milk. The high sequence coverage for BLG, with all but one sequence digested by trypsin, is an indication for the presence of intact BLG in human milk. This is supported by the findings of Zhu et al., who, although with a lower sequence coverage, also identified BLG in human milk. In addition, several other studies reported the identification of intact BLG in human milk using immunochemical analysis [2,26,27]. The presence of intact BLG may be due to its relatively small size and its high resistance to pepsin digestion, as previously shown by the presence of intact BLG in jejunum samples [28]. From our results, it appears that especially proteinaceous molecules from bovine milk end up in human milk. This might relate to the high consumption rate of dairy products in the Netherlands, considering that 23% of the average dietary protein intake originates from milk and dairy products [29]. In line with this, the highly abundant bovine milk serum proteins (BLG and BSA) were identified with the highest sequence coverage. Bovine ALA, another major milk serum protein, was identified with only two peptide sequences. This low sequence coverage was expected because of its high digestibility and high homology with human ALA. This high homology reduces the number of unique peptides that can be identified from this protein. Besides the bovine peptide sequences, peptide sequences of cat and horse serum albumin were identified and confirmed by the analysis of synthesized peptides. No relation was found between the presence of cat serum albumin peptides in milk and the ownership of a cat as pet by the respective mothers. Nevertheless, it is known that mammalian serum albumins are present in animal dander and exposure to this is not limited to direct contact [30,31]. The serum albumins of cat or horse could end up in the human digestive system by ingestion or inhalation and could subsequently be transferred to the milk. Whether these proteins are present as intact proteins or in large fragments remains unclear because of the relatively low sequence coverage that was found. Several other studies reported the detection of other dietary allergen proteins in human milk, such as egg ovalbumin, peanut allergen, and wheat gliadin [6,9,13]. Peptide sequences of these three proteins were initially detected in the current study but were filtered out due to a low PSM score or to not being confirmed by analysis of synthesized peptides. This could still mean that these proteins are present in human milk but in too low concentrations for positive identification.

Several previous studies have investigated a possible difference in nonhuman proteins between milk from allergic (maternal history of atopic diseases) and nonallergic mothers. Høst et al. [26] and more recently Matangkasombut et al. [32] did not find a difference in BLG levels in milk between the two groups. Another study, also investigating BLG levels, found BLG in the milk of all allergic subjects involved and not in the milk of all nonallergic subjects [27]. Sorva et al. found that BLG levels in milk of allergic and nonallergic mothers were similar after 24 h on a milk-free diet [33]. Nevertheless, the levels of BLG tended to be higher in milk from allergic mothers one hour after consumption of 400 mL of bovine milk. Surprisingly, the current study shows a significant difference concerning peptide sequences originating from BLG and BSA. A similar finding has not been reported before. The difference between our results and the aforementioned studies can possibly be explained by the characteristics of the allergic subjects and a difference in methodology. The current study has, for example, a strict selection on both HDM-specific IgE and allergy symptoms, whereas previous studies did not elaborate on the definition they used for atopy or allergy, and in some cases, the selection of allergic subjects was based on symptoms only. In addition, immunochemical analyses have shown to be influenced by cross-reactivity between human and bovine proteins, which make them less reliable than LC-MS/MS analysis [34,35]. For the current study, only an indication of the frequency of consumption for milk and milk products was available (Table 1). Nevertheless, it seems that the allergic mothers even consumed less milk or milk products when compared to the nonallergic mothers. Therefore, our findings indicate that there is a difference between the allergic and nonallergic mothers when it comes to the transfer of bovine proteinaceous molecules from the intestinal tract to the mammary gland.

From the intestinal tract to the blood, proteins can be absorbed by both paracellular and transcellular pathways. In reviewing the literature, Reitsma et al. suggested that a difference in intestinal absorption of proteins between non-sensitized and sensitized persons can take place by both pathways [10]. Which of these two is involved in the transfer of dietary proteins into human milk is not known. One option for a transcellular pathway concerns transport of intact antigens with specific IgE via the CD23 receptor [36]. With regard to the current study, this would suggest an increased level of BLG-specific IgE in HDM allergic mothers, which seems unlikely and has not been mentioned in literature. Another transcellular pathway is through enterocytes and involves degradation of the protein in lysosomes [37]. A recent study using CaCo-2 cell monolayers showed that casein fragments survive transfer by this pathway but that BLG seems to be completely degraded [38]. Therefore, this pathway seems unlikely. A third transcellular pathway is via M cells, and it has been suggested that BLG can be transferred through this pathway without degradation [39]. Nevertheless, there is no evidence that transport through these pathways is increased in allergic mothers. A prerequisite for the uptake of proteinaceous molecules through the paracellular pathway is an impaired intestinal barrier. Reitsma et al. pointed out that sensitized persons have an increased level of mast cells that release IgE-induced tryptase [10]. The tryptase affects the tight junctions in the intestinal barrier, leading to an increased permeability, which could allow the passage of proteinaceous molecules. Nevertheless, this pathway is linked to food hypersensitivity reactions where the location of sensitization is in the intestinal tract itself. In patients with HDM allergy, sensitization occurs primarily in the respiratory tract. However, Calderón et al. suggested that HDM sensitization can be systemic and could cause reactions in other parts of the body [40]. Such a lung–gut cross-talk is plausible, considering the evidence showing that the mucosal immune system can be considered as a system-wide organ [41]. In reviewing the literature, Zhu et al. suggested, in line with this hypothesis, the role of a thymic stromal lymphopoietin-mediated pathway that is induced by HDM allergen sensitization, which might promote the breakdown of the epithelial barrier in the intestinal tract [42].

Several other factors could be involved in the disruption of the intestinal barrier. Firstly, it has been shown that, independent of atopy, asthma can be associated with an increased intestinal barrier permeability [14,43]. In the current study, seven out of 10 of the allergic mothers reported asthma (Table 1). Secondly, Tulic et al. showed that HDM is often present in the gut and that its cysteine protease Der p 1 causes disruption of the epithelial barrier [44]. This disruption appeared to be similar for HDM-sensitized and HDM-non-sensitized subjects. Nevertheless, due to an inflammatory response, it might be possible that recovery of the intestinal barrier dysfunction is delayed or incomplete in allergic subjects. This might then explain the permeability of the intestinal barrier in the allergic subjects in the current study, the majority of whom have HDM allergy. it should be noted that the majority of the in vivo studies on intestinal permeability make use of small inert molecules and their passage through the intestine. it has been shown that an increased transfer of these molecules through the intestinal barrier does not necessarily correlate with the transfer of antigens [45]. In addition, a previous study showed, using ELISA, that levels of BLG in human milk were not related to intestinal barrier permeability [33]. Therefore, more research is needed to elucidate whether the increased barrier permeability caused by these factors indeed leads to an increased passage of proteinaceous molecules.

After passage through the intestinal barrier, it is expected that the nonhuman proteinaceous molecules enter the blood stream and are subsequently transferred through the mammary epithelium into the alveolar lumen. This transfer seems to take place through a one-way transcytotic pathway. Monks et al. showed the role of this pathway in the transfer of extracellular serum albumin in mice and suggested that this is the same pathway that is involved in the transfer of IgA [12]. After transfer to the milk, the nonhuman proteinaceous molecules end up in the digestive system of the infant. Worth noting is that Hettinga et al., in analyzing the same data but focusing on human proteins, found significantly higher levels of several protease inhibitors in human milk from allergic mothers [19]. These protease inhibitors (cystatin C, inter-alpha-trypsin inhibitors, and serine-protease inhibitors) are potentially active against enzymes that hydrolyse BLG. Consequently, the human milk composition of allergic mothers might reduce the hydrolysis of BLG in the infant’s intestinal tract.

Since the current study does not include absolute quantification, the exact level of these molecules in human milk remains unclear. Regardless of their level, it is known that bovine milk proteins in human milk can have an effect on the infant. Several reported cases described non-IgE-mediated food protein-induced enterocolitis syndrome caused by bovine milk proteins in exclusively breastfed infants [46,47]. In all these cases, the infant had a positive family history for atopy and clinical manifestations were resolved after the mother strictly eliminated cow’s milk from her diet. It remains unclear whether nonhuman proteinaceous molecules in human milk can have an effect on the development of the immune system of the breastfed infant beyond causing allergic symptoms. Verhasselt et al. showed, using a mouse model, that antigen transfer through breastmilk induced tolerance and protection from allergic asthma [48]. Translating this to BLG, it is known that BLG-derived peptides can be HLA-DRB1-restricting, a characteristic that might support oral tolerance development [49]. In line with this, Peters et al. showed recently that early introduction of cow’s milk was associated with a reduced risk of cow’s milk allergy [50]. The presence of higher levels of BLG or its derived peptides in human milk of allergic mothers might therefore have a protective effect on further allergy development. Nevertheless, evidence remains speculative and a direct relation needs to be investigated.

Interestingly, BSA peptide sequences were found in significantly lower levels in milk from allergic mothers. Previous research with rats showed that intact BSA can pass the intestinal epithelium [51]. Nevertheless, the difference found between the two groups is difficult to interpret. The most likely but speculative explanation is a specific pathway that is activated in healthy mothers but that is negatively regulated in allergic mothers.

## 5. Conclusions

In conclusion, in the present study, a significant difference in levels of nonhuman proteinaceous molecules in human milk of allergic and nonallergic mothers has been observed. Sequences from BLG appeared in higher levels and sequences from BSA appeared in lower levels in milk from allergic mothers when compared to milk from nonallergic mothers. These findings suggest that there is a difference in transfer of proteinaceous molecules through the intestinal barrier of allergic mothers, allowing dietary proteins to enter the bloodstream and ultimately the milk. This study has raised important questions about the role that these proteinaceous molecules might play in the development of the immune system of infants. 

## Figures and Tables

**Figure 1 nutrients-12-01169-f001:**
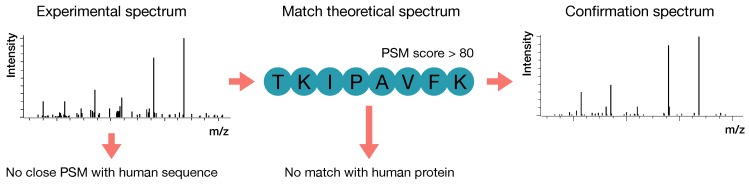
Schematic overview of the workflow used for confirmation of the identified nonhuman peptide sequences. After LC-MS/MS analysis, experimental MSMS spectra that matched theoretical nonhuman peptide sequences were selected when there was no close peptide sequence match (PSM) with a human peptide sequence. Spectra with a PSM score < 80 or a full match with the human proteome were removed. The final remaining spectra were confirmed with retention time and MSMS spectra of bovine milk, pure proteins, or synthetically acquired peptides.

**Figure 2 nutrients-12-01169-f002:**
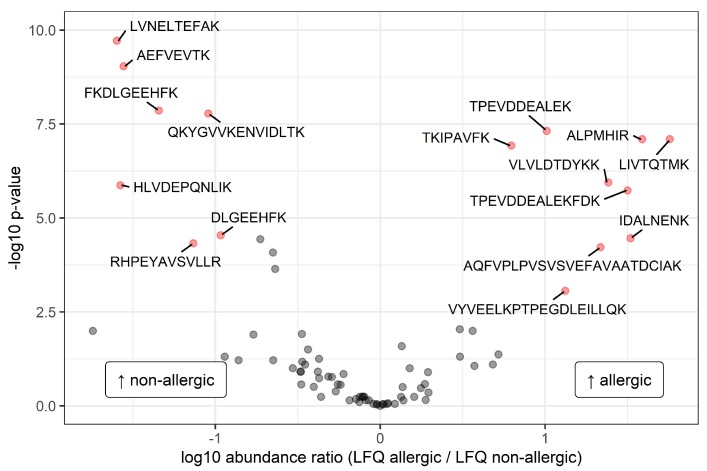
Volcano plot with the ratios of the group means of the ^10^log transformed label-free quantification (LFQ) intensities of the identified peptide sequences. Significantly different peptides (false discovery rate < 0.01 and difference between groups > ±0.75) are represented by filled red circles and labelled with the corresponding amino acid sequence. On the right side of the plot, the peptides with a higher level in allergic mothers are presented, and on the left side, the peptides with a higher level in nonallergic mothers are presented.

**Figure 3 nutrients-12-01169-f003:**
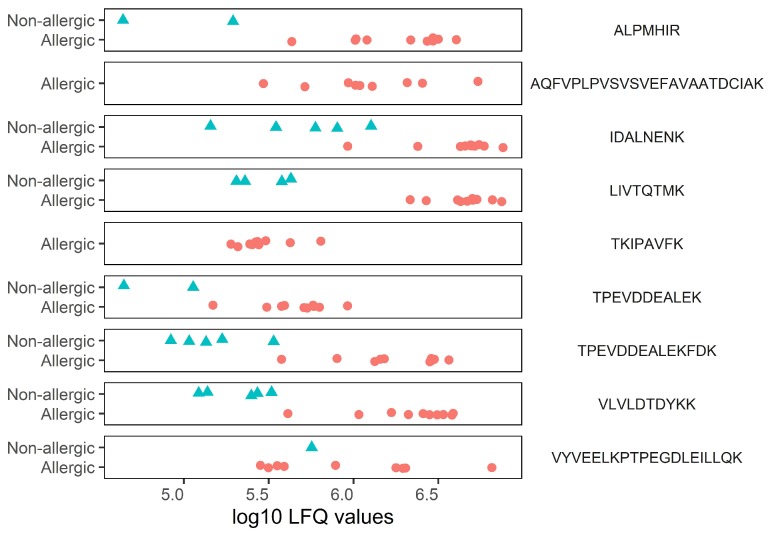
Categorical scatterplot showing non-imputed, ^10^log transformed LFQ intensities of peptide sequences that were found in significantly higher levels in milk from allergic mothers. Allergic mothers are represented by red circles, and nonallergic mothers are represented by blue triangles.

**Table 1 nutrients-12-01169-t001:** Details on the mothers included in the sample collection, with allergy status, Der p IgE Rast-class of the allergic mothers, presence of a cat as pet, and consumption of milk and dairy products.

Characteristic	Type	Nonallergic	Allergic
House dust mite allergy	Self report	0	7
	Doctor diagnosed	0	7
House dust allergy	Self report	0	8
	Doctor diagnosed	0	6
Allergic to pets	Self report	0	9
	Doctor diagnosed	0	8
Asthma	Self report	0	7
	Doctor diagnosed	0	7
House dust mite Der p IgE (Rast-class)	Class 3	NA ^a^	4
	Class 4	NA ^a^	5
	Class 5	NA ^a^	1
Rhinitis/hay fever	Self report	0	9
Cat as pet in the household	Presence	3	3
Consumption of milk during lactation	Not at all	2	3
	1–3 × a month	0	1
	1 × a week	0	0
	2–4 × a week	0	3
	More than 4 × a week	0	0
	1 × a day	1	1
	Multiple times a day	7	2
Consumption of milk products during lactation	Not at all	2	0
	1–3 × a month	0	0
	1 × a week	0	0
	2–4 × a week	1	0
	More than 4 × a week	0	1
	1 × a day	4	4
	Multiple times a day	3	5

^a^ NA = Data not available.

**Table 2 nutrients-12-01169-t002:** All identified nonhuman peptide sequences that were assigned to bovine proteins, with the corresponding UniProt protein id, name of the leading protein, and in silico digestion mode. Per group of allergic and nonallergic mothers, the number of samples in which the peptide sequence was identified is listed.

Sequence	Leading Proteins	Protein Names	Allergic	Nonallergic	Digestion
ALPMHIR ^a^	B5B0D4	Beta-lactoglobulin	10	2	trypsin
IDALNENK ^a^	B5B0D4	Beta-lactoglobulin	10	5	trypsin
LIVTQTMK ^a^	B5B0D4	Beta-lactoglobulin	10	4	trypsin
LSFNPTQLEEQCHI ^b^	B5B0D4	Beta-lactoglobulin	10	6	trypsin
TKIPAVFK ^a^	B5B0D4	Beta-lactoglobulin	10	0	trypsin
TPEVDDEALEK ^a^	B5B0D4	Beta-lactoglobulin	10	2	trypsin
TPEVDDEALEKFDK ^a^	B5B0D4	Beta-lactoglobulin	10	5	trypsin
VLVLDTDYKK ^a^	B5B0D4	Beta-lactoglobulin	10	5	trypsin
VYVEELKPTPEGDLEILLQK ^a^	B5B0D4	Beta-lactoglobulin	9	1	trypsin
WENDECAQK ^b^	B5B0D4	Beta-lactoglobulin	9	1	trypsin
WENDECAQKK ^b^	B5B0D4	Beta-lactoglobulin	4	0	trypsin
SLAMAASDISLLDAQSAPLR ^b^	B5B0D4	Beta-lactoglobulin	6	0	semi-specific
HHIELRWK	E1BFN5	Uncharacterized protein	9	8	trypsin
QKYGVVKENVIDLTK	E1BJP1, G3MZU3	Uncharacterized proteins	0	9	semi-specific
EKESLGWQK	E1BKT9	Desmoplakin	0	2	unspecific
EHLYQENQYLEQENTQ	E1BMB1	Ninein	0	6	unspecific
QEELENRTSETNTPQGNQEY	E1BMB1	Ninein	8	3	unspecific
HEQGMDQDKN	F1MV51	APC, WNT signalling pathway regulator	10	10	unspecific
SSLSDIDQENNNNK	F1MV51	APC, WNT signalling pathway regulator	2	3	unspecific
TLQIAEIKDNSGPRSNED	F1MV51	APC, WNT signalling pathway regulator	0	2	unspecific
QNLAFVSMLNDIAAP	F1N647	Fatty acid synthase	0	1	unspecific
IQQNSSTTEKI	F2FB38	Mucin-16	6	9	unspecific
KFNITDTLMQ	F2FB38	Mucin-16	0	1	unspecific
LDQWLCEKL ^b^	P00711	Alpha-lactalbumin	4	0	trypsin
NICNISCDKFLDD	P00711	Alpha-lactalbumin	0	1	unspecific
EKVNELSK ^a^	P02662	Alpha-S1-casein	7	1	trypsin
FFVAPFPEVFGK ^a^	P02662	Alpha-S1-casein	2	3	trypsin
HIQKEDVPSER ^a^	P02662	Alpha-S1-casein	10	8	trypsin
HQGLPQEVLNENLLR ^a^	P02662	Alpha-S1-casein	5	8	trypsin
YLGYLEQLLR ^a^	P02662	Alpha-S1-casein	2	5	trypsin
SCQAQPTTMAR ^b^	P02668	Kappa-casein	9	3	trypsin
AEFVEVTK ^a^	P02769	Serum albumin	7	10	trypsin
DAFLGSFLYEYSR ^a^	P02769	Serum albumin	6	4	trypsin
DLGEEHFK ^b^	P02769	Serum albumin	0	9	trypsin
DTHKSEIAHR ^a^	P02769	Serum albumin	0	10	trypsin
DVCKNYQEAK ^b^	P02769	Serum albumin	10	10	trypsin
FKDLGEEHFK ^a^	P02769	Serum albumin	10	10	trypsin
HLVDEPQNLIK ^a^	P02769	Serum albumin	4	9	trypsin
LVNELTEFAK ^a^	P02769	Serum albumin	7	10	trypsin
QNCDQFEK ^b^	P02769	Serum albumin	0	5	trypsin
RHPEYAVSVLLR ^a^	P02769	Serum albumin	7	10	trypsin
SLHTLFGDELCK ^b^	P02769	Serum albumin	1	8	trypsin
TCVADESHAGCEK ^b^	P02769	Serum albumin	2	7	trypsin
GKYLYEIAR	P02769	Serum albumin	9	10	semi-specific
KQTALVELLK ^b^	P02769	Serum albumin	2	5	unspecific
IKVMNDLSPKSNLR	P07353	Interferon gamma	2	1	semi-specific
DLKLVEQQNPK	P08037	Beta-1,4-galactosyltransferase 1	0	2	semi-specific
AQFVPLPVSVSVEFAVAATDCIAK ^b^	P12763	Alpha-2-HS-glycoprotein	9	0	trypsin
VNLLVDRQWQAVRNR	P15396	Ectonucleotide pyrophosphatase	10	10	trypsin
KLLNNITNDLR	P21758	Macrophage scavenger receptor	4	0	unspecific
NLLFNDNTECLAK ^b^	P24627	Lactotransferrin	4	1	trypsin
NKHSNLIESQENSK	P31098, P31096	Osteopontin-K, Osteopontin	9	7	trypsin
NVTRQAYWQIHMDQ	P80209	Cathepsin D	0	3	unspecific
NGNNPNCCMNQK	P80457	Xanthine dehydrogenase/oxidase	1	0	semi-specific
EKQLPNGDWPQENISGVFNKSCA	P84466	Lanosterol synthase	5	3	unspecific
VSITCSGSSSNIGR ^b^	Q1RMN8	Immunoglobulin light chain	8	5	trypsin
CASFRENVLR ^b^	Q29443	Serotransferrin	10	10	trypsin
QMERALLENE	Q2HJ49	Moesin	0	3	semi-specific
NGEGQVLFETEISR	Q2TBX4	Heat shock 70 kDa protein 13	3	8	trypsin
NIIKSGSDEVQ	Q2UVX4	Complement C3	1	0	unspecific
VALNKLK	Q58D55	Beta-galactosidase	2	0	trypsin
VYVEQLKPTPEGDLEILLQK	Q9BDG3	Beta lactoglobulin D	1	0	trypsin

^a^ confirmed by analysis of digested pure protein; ^b^ confirmed by analysis of bovine milk serum proteins

**Table 3 nutrients-12-01169-t003:** All identified nonhuman peptide sequences that were assigned to non-bovine allergens with the corresponding UniProt protein id, leading organism, and in silico digestion mode. Per group of allergic and nonallergic mothers, the number of samples in which the peptide sequence was identified is listed.

Sequence	Leading Proteins	Leading Organisms or LCA ^b^	Allergic	Nonallergic	Digestion
QNWASLQPYKKL	Q08169, A0A0M9A8V0, I1VC83, A0A2A3EHG0, Q95PD7, A0A0L7RCK4, A0A310SIY9	Apidae (family) (bees)	1	2	semi-specific
RPSHQQPR	P43237, N1NEW2	Arachis (genus) (legumes)	6	4	trypsin
MQDQLDQVQK	Q8MUF6, Q9BMM8, A0A1B2YLJ8	Astigmatina (cohort) (mites)	1	5	unspecific
KELKKKVEADGEND	A0A2V1CGL9	*Cadophora* sp. DSE1049	6	4	unspecific
QIANSDEVEKI	Q24702	*Dictyocaulus viviparus*	3	6	unspecific
KCAADESAENCDK	P35747	*Equus caballus*	7	3	trypsin
LVNEVTEFAKK ^a^	P35747	*Equus caballus*	10	8	trypsin
KEPERNECFLQHK ^a^	P49064	*Felis catus*	5	8	trypsin
PCFSALQVDETYVPK	P49064	*Felis catus*	1	0	trypsin
YICENQDSISTK	P49064	*Felis catus*	0	5	trypsin
SALQVDETYVPK	P49064	*Felis catus*	3	4	semi-specific
KEQVARFTAGTNPK	A9QQ26	*Lycosa singoriensis*	10	10	trypsin
EQVQELR	A0A1L8GUE3, A0A3Q0GE46, A0A151P804	Tetrapoda (superclass) (4-limbed vertebrates)	2	2	trypsin
QQQTLQQILQQQ	P04723	*Triticum aestivum*	10	10	unspecific
QVLQQSSYQQLQQ	P04723	*Triticum aestivum*	0	2	unspecific
QFKPEEMTNIIK	P35083, A4KA55	*Zea mays*	8	4	semi-specific

^a^ confirmed by analysis of acquired synthesized peptide; ^b^ last common ancestor (LCA) in case of multiple leading organisms.

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
