# Peer review of "Maternal Allergy and the Presence of Nonhuman Proteinaceous Molecules in Human Milk"

_nutrients, 2020, doi:10.3390/nu12041169_

Round 1

Reviewer 1 Report

Very good paper, well written. I have no major concerns, only a couple of minor suggestions.

The initial study had a large number of participants but only 10 women considered allergic and 10 considered non-allergic were selected for this study. Maybe a bit of a rationale as to why only this number was chosen and whether they were chosen randomly.

Maybe include what the abbreviation "NA" is in full somewhere. I did not find it.

I found it interesting that BLG was preferentially present in allergic mothers when this was not the case with other bovine proteins. It was discussed in the discussion and the answer is not known but interesting none the less.

Author Response

Dear sir/madam,

Thank you for your time and effort in reviewing our manuscript.

Comment 1: The initial study had a large number of participants but only 10 women considered allergic and 10 considered non-allergic were selected for this study. Maybe a bit of a rationale as to why only this number was chosen and whether they were chosen randomly.

Response: Thank you for pointing this out. We had briefly described the power calculation in line 70-71, but we have elaborated more on it in the new version of the manuscript: “The number of mothers included is based on a power calculation, aiming at finding a 5-fold difference, as detailed in Hettinga et al. [19].”

Comment 2: Maybe include what the abbreviation "NA" is in full somewhere. I did not find it.

Response: This could be referring to line 204-205, the NAs there were a mistake. Due to a typing error in the LaTeX code, the numbers did not show up here and instead, NA was shown. The numbers that should be shown are 15 and 11, respectively. We have changed this in the final version.

Regarding the NA entries in Table 1. We have added a footnote to the table, explaining that NA means “Data not available”.

Reviewer 2 Report

This research paper reports on the relationship between maternal allergy and the presence of non-human proteinaceous molecules in human milk. The authors employ sound methodology and the figures are detailed and support the conclusions well. This study demonstrates that there are significant differences between the levels of molecules such as BLG detected in the milk of allergic mothers compared to those who are non-allergic. The authors carefully discuss paracellular and transcellular transport mechanisms that could account for the transfer of such molecules from the digestive tract to the mammary gland. 

It is a strength of the study that allergic mothers were selected based on defined immunological characteristics such as presence of HDM specific IgE. The authors demonstrate cautious interpretation of the data and emphasize that the nature of the study is mostly correlational. The data presented in this article has implications for future research in that the relationship between BLG in allergic mothers and a possible protective against the development of allergies in infants. 

 Minor editorial changes are in order:

  1. Lines 204-205: the statement is contradictory, probably the authors meant to say NA sequences were not identified in non-allergic mothers.
  2. Line 203: the subheading would read better as "Differences between allergic and non-allergic mothers"

Author Response

Dear sir/madam,

Thank you for your time and effort in reviewing our manuscript.

Comment 1: Lines 204-205: the statement is contradictory, probably the authors meant to say NA sequences were not identified in non-allergic mothers.

Response: Thank you for pointing this out. The NAs there were a mistake. Due to a typing error in the LaTeX code, the numbers did not show up here and instead, NA was shown. The numbers that should be shown here are 15 and 11, respectively. We have changed this in the new version.

Comment 2: Line 203: the subheading would read better as "Differences between allergic and non-allergic mothers"

Response: Agree. We have changed this accordingly in the new version of the manuscript.